# Structural insights into the gating of DNA passage by the topoisomerase II DNA-gate

Shin-Fu Chen [1], Nan-Lan Huang [2,3], Jung-Hsin Lin[2], Chyuan-Chuan Wu [1,4], Ying-Ren Wang [1], Yu-Jen Yu[1], Michael K. Gilson [3] & Nei-Li Chan [1,5,6]

Type IIA topoisomerases (Top2s) manipulate the handedness of DNA crossovers by introducing a transient and protein-linked double-strand break in one DNA duplex, termed the DNA-gate, whose opening allows another DNA segment to be transported through to change the DNA topology. Despite the central importance of this gate-opening event to Top2 function, the DNA-gate in all reported structures of Top2-DNA complexes is in the closed state. Here we present the crystal structure of a human Top2 DNA-gate in an open conformation, which not only reveals structural characteristics of its DNA-conducting path, but also uncovers unexpected yet functionally significant conformational changes associated with gate-opening. This structure further implicates Top2's preference for a left-handed DNA braid and allows the construction of a model representing the initial entry of another DNA duplex into the DNA-gate. Steered molecular dynamics calculations suggests the Top2-catalyzed DNA passage may be achieved by a rocker-switch-type movement of the DNA-gate.

[1] Institute of Biochemistry and Molecular Biology, College of Medicine, National Taiwan University, Taipei 10051, Taiwan. [2] Research Center for Applied Sciences and Institute of Biomedical Sciences, Academia Sinica, Taipei 11529, Taiwan. [3] Skaggs School of Pharmacy and Pharmaceutical Sciences, University of California, San Diego, La Jolla 92093 CA, USA. [4] Institute of Molecular Biology, Academia Sinica, Taipei 11529, Taiwan. [5] Institute of Biochemistry, National Chung Hsing University, Taichung 40227, Taiwan. [6] Scientific Research Division, National Synchrotron Radiation Research Center, Hsinchu 30076, Taiwan. Correspondence and requests for materials should be addressed to M.K.G. (email: mgilson@ucsd.edu) or to N.-L.C. (email: nlchan@ntu.edu.tw)

Type IIA topoisomerases (Top2s) are essential DNA-manipulating enzymes ubiquitous in eukaryotes and bacteria[1–3]. These enzymes exploit protein conformational changes driven by ATP binding and hydrolysis to direct the crossing of two DNA duplexes, leading to topological inversion of DNA crossovers[2,4]. This unique duplex DNA passage activity allows the resolution of intra- and intermolecular DNA entanglements that arise from cellular DNA transactions, including replication, transcription, chromosome segregation, and recombination[1,4,5].

The Top2-mediated DNA topological transformation requires the temporary creation of a DNA double-strand break (DSB) on one DNA segment (the G-segment), so that another duplex strand (the T-segment) can be transported through[2,3,5]. To produce this essential DSB, the twofold symmetric Top2 first associates with the G-segment through a positively charged groove formed by the WHD, tower, and TOPRIM domains[6–8]. A pair of catalytic tyrosines from the WHD domains then initiates a transesterification reaction by attacking at two 4-bp staggered phosphodiester bonds on opposite DNA strands, giving rise to the "Top2 cleavage complex" which harbors the so-called "DNA-gate" featuring a cleaved G-segment with the enzyme covalently attached to the 5′-ends via a phosphotyrosyl linkage[9–12] (Fig. 1a). Next, the two N-terminal ATPase domains, which together function as an ATP-operated gate (the N-gate), undergo a nucleotide-dependent closure to capture and drive the T-segment through the DNA-gate[13–15]. The enzyme is reset for its next catalytic cycle by subsequent steps[3,4,6], which include the release of the T-segment through the C-gate formed by the helical domain appended to the coiled-coil linker, religation of the cleaved G-segment by reversal of the transesterification reaction, and re-opening of the N-gate dimer interface upon ATP hydrolysis.

Because directional transport of the T-segment is achieved via coordinated opening and closure of the N-gate, DNA-gate, and C-gate, a full understanding of the catalytic mechanism of Top2 requires structural characterization of these three gates in distinct conformational states. Crystallographic studies have revealed the structural details of the N-gate in its open[16] and nucleotide-bound, closed[17–19] forms. The C-gate has also been resolved in both open[8,20] and closed[11,12] conformation. In contrast, whereas multiple crystal structures are available for the closed DNA-gate[8,11,12,21–23], opening of the DNA-gate has never been directly visualized. Consequently, outstanding issues regarding the operation of the DNA-gate remain poorly defined, including the architectural and surface features of the T-segment-conducting path, the tertiary and quaternary structural changes associated with gate-opening, and the structural changes of the enzyme-linked DNA cohesive ends upon their detachment from one another. Thus, a step-by-step description of how the T-segment is transported through the DNA-gate is not yet available. In addition, various clinically active anticancer drugs and antibacterials act by targeting the Top2 DNA-gate to produce cytotoxic DNA lesions[3,24,25]. Obtaining a more complete picture of the conformational landscape of the DNA-gate will contribute to the development new Top2-targeting agents.

We report herein the high-resolution view of the opening of the Top2 DNA-gate, which directly mediates the resolution of topological strand crossings. This structure not only reveals the formation and structural features of the T-segment-conducting path, but also uncovers unexpected, functionally relevant, conformational changes that accompany the closed-to-open transition. Moreover, using this structure as a starting point, we simulated the passage of the T-segment through the DNA-gate by steered molecular dynamics (SMD) simulations, to gain further insight into this central yet elusive step in Top2 catalysis.

## Results

**Top2 cleavage complex structure reveals opening of DNA-gate.** Crystallization has been recognized as an effective approach for sampling energetically accessible conformational states of proteins and protein complexes[26,27]. The structures of many allosteric proteins obtained from different crystal forms have been interpreted to represent distinct functional states of the protein, and hence provide valuable insights into the mechanistic basis of protein conformational switching and protein allostery. We determined the structure of the binary complex formed by the DNA-binding and cleavage core of human topoisomerase 2β (hTop2β$^{core}$, Fig. 1b) and a 20-bp duplex DNA in a new crystal form, at 2.75-Å resolution, by molecular replacement (Table 1). The asymmetric unit contains one protein subunit bound to a 9-bp DNA duplex, about half the length of the palindromic DNA used for the preparation of the dimeric Top2 cleavage complex (Fig. 1b). A pair of monomers from two neighboring asymmetric units were found to interact via their C-terminal primary dimerization domains (the C-gate region), thus a functionally relevant homodimer can be readily constructed by applying a rotation about the crystallographic twofold axis (Fig. 1c and Supplementary Figure 1).

In all previously reported structures of Top2 cleavage complexes[11,12,21–23], the DNA-gate regions are in the closed conformation with no discernable cavity to accommodate the passage of the T-segment DNA (Fig. 1d, left). In contrast, the current structure adopts a quaternary conformation whose dimeric architecture is held solely by the interactions formed between the two C-gate regions without any direct or indirect contacts being observed between the two WHD domains (Fig. 1c, d). As the G-segment-binding grooves of the two monomers have moved away from each other, the continuous G-segment-binding surface that spans across the two WHD domains in the closed structure is disrupted, and a funnel-shaped trough is revealed between the two halves of the DNA-gate (Fig. 1c), effectively opening an entryway for the T-segment. This conformation may thus be taken as a crystallographic snapshot of the DNA-gate in an "open" state during the Top2 functional cycle.

If this snapshot does represent a Top2 cleavage complex in an open conformation, it should include the characteristic phosphotyrosyl bonds between the enzyme's catalytic tyrosine residues and the 5′-phosphate groups of the cleaved G-segment DNA. As the two cohesive ends that are expected to result from Top2-mediated cleavage of G-segment are not visible in the electron density map (Supplementary Figure 2), we validated the formation of the Top2 cleavage complex via two additional studies. First, the hTop2β$^{core}$-DNA complex retrieved by dissolving an aliquot of the crystals was subjected to SDS-PAGE analysis (Fig. 2d). The appearance of a protein species with electrophoretic mobility matching that of the standard covalent hTop2β$^{core}$-DNA adduct, and slower than that of the DNA-free Top2, indicates the existence of the cleavage complex in the crystal, consistent with the mechanistic relevance of the structure. Second, we sought to exclude the possibility that the 9-bp DNA fragment seen in the asymmetric unit reflects binding of Top2 to a single-stranded hairpin loop, instead of a cleaved double-stranded fragment (Figs. 1b and 2a, c). This was done by generating new crystals, this time using a substrate G-segment DNA in which the thymidine nucleotides were replaced by 5-iodouridine, to re-determine the structure reported here, as well as a previously reported cleavage complex[23]. Because iodine is substantially more electron-rich than the 5-methyl group of thymine, the use of 5-iodouridine allowed the positions of thymine nucleotides (Fig. 2a) to be unambiguously recognized in the difference electron density map. Here, if the DNA bound in the asymmetric unit is a single-stranded hairpin, the I-labels should exhibit an altered pattern and be observed on both sides of the stem (Fig. 2c).

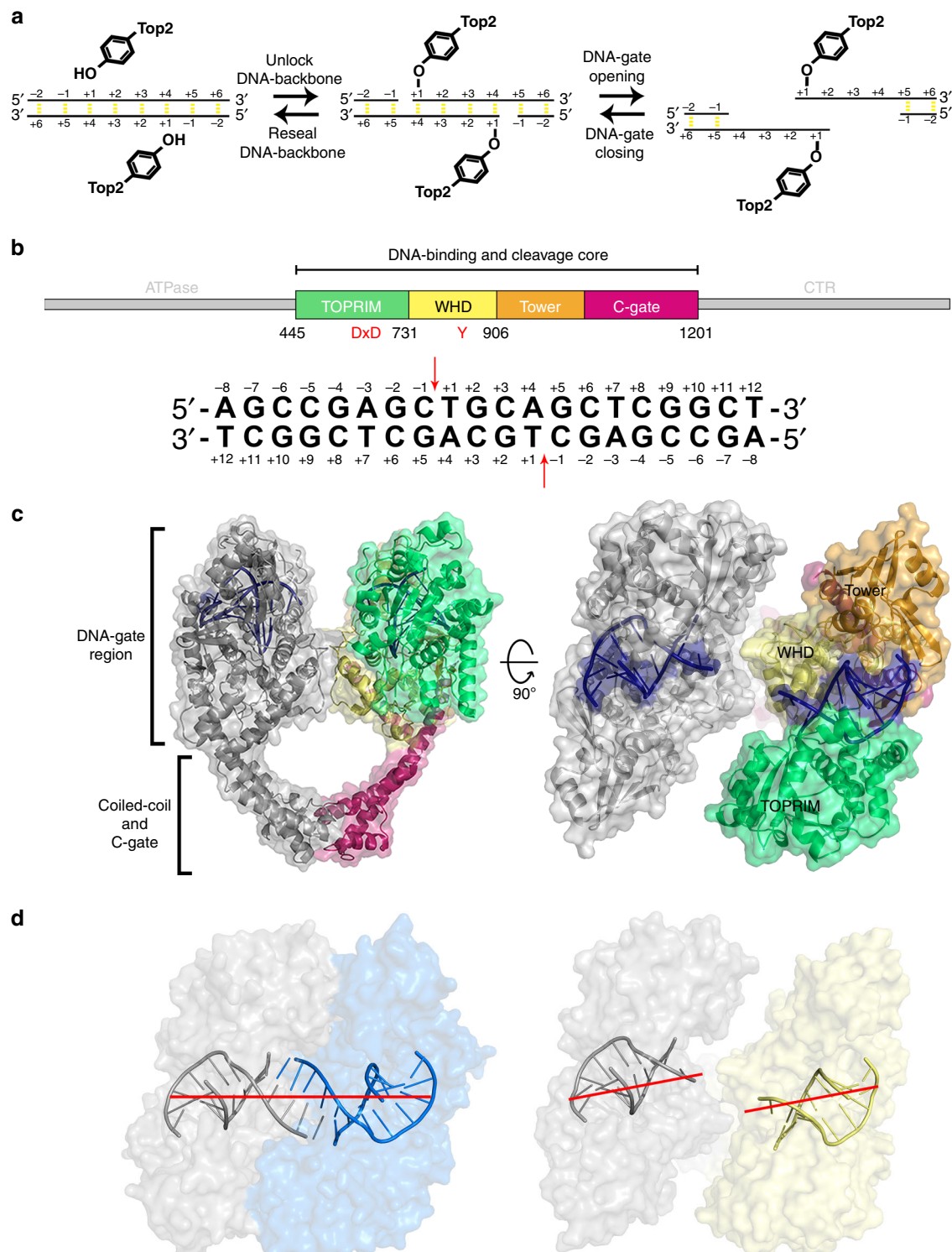

**Fig. 1** Structure of Top2 cleavage complex showing opening of the DNA-gate. **a** Opening of the Top2-mediated DNA-gate involves two reversible steps. First, Top2 catalyzes transesterification reactions between a pair of catalytic tyrosines and two opposing phosphodiester bonds 4-bp apart, which unlocks the backbones of the G-segment. During this process, the catalytic tyrosine is linked to the 5′-phosphate of the +1 nucleotide via a phosphotyrosyl linkage, leaving a free 3′-OH at the −1 position. Next, the enzyme pulls the two halves of the cleaved G-segment apart to form a gate wide enough for the T-segment to go through. No structural information was available regarding the opening of the DNA-gate until now. **b** Top panel: linear domain organization of eukaryotic Top2. The middle fragment, corresponding to Top2's DNA-binding and cleavage core, termed hTop2core, was used in this study. Bottom panel: the duplex DNA used for crystallization with the cleavage sites indicated by arrows. **c** Orthogonal views of the new hTop2βcore–DNA cleavage complex structure. DNA is in blue, one Top2 monomer is in gray, and the other is colored following the scheme in **b**. **d** The G-segment in the new structure (right) has been cleaved, and the two fragments are apart from each other. In contrast, in the closed conformation (left, PDBid: 3L4K) the cleaved DNA ends remain tethered by Watson–Crick base pairing. The red lines indicate the helical axes of DNA duplexes

**Table 1 Data collection and refinement statistics**

| | hTop2β$^{core}$-DNA-binary complex | hTop2β$^{core}$-DNA$^I$-binary complex | hTop2β$^{core}$-DNA$^I$-etoposide ternary complex |
|---|---|---|---|
| PDB ID code | 5ZEN | 5ZQF | 5ZRF |
| **Data collection** | | | |
| Space group | $P3_221$ | $P3_221$ | $P2_1$ |
| Cell dimensions | | | |
| $\quad a, b, c$ (Å) | 95.77, 95.77, 231.19 | 95, 95, 231.5 | 80.12, 176.46, 94.4 |
| $\quad \alpha, \beta, \gamma$ (°) | 90, 90, 120 | 90, 90, 120 | 90, 111.4, 120 |
| Resolution (Å) | 20.00–2.75 | 20.00–3.87 | 20.00–2.3 |
| | (2.85–2.75) | (4.01–3.87) | (2.38–2.3) |
| Observed reflections | 218,295 | 78,409 | 278,909 |
| Unique reflections | 32,586 (3182) | 11,767 (1158) | 99,792 (9152) |
| Redundancy | 6.4 (6.7) | 6.7 (6.8) | 2.9 (2.9) |
| Completeness (%) | 99.6 (99.7) | 99.2 (100) | 92.2 (85.2) |
| $I/\sigma I$ | 33.3 (3.8) | 14.1 (3.09) | 18.4 (2.33) |
| $R_{sym}$[a] | 0.05 (0.48) | 0.15 (0.7) | 0.05 (0.47) |
| $CC_{1/2}$ | 0.98 (0.91) | 0.97 (0.87) | 0.93 (0.73) |
| **Refinement** | | | |
| Wison B-factor | 68.09 | 105.8 | 37.53 |
| $R_{crys}$[b] (%) | 0.22 (0.29) | 0.20 (0.27) | 0.19 (0.24) |
| $R_{free}$[b] (%) | 0.26 (0.33) | 0.25 (0.32) | 0.23 (0.31) |
| R.M.S. deviations | | | |
| $\quad$ Bond lengths (Å) | 0.006 | 0.004 | 0.006 |
| $\quad$ Bond angles (°) | 0.84 | 0.54 | 0.79 |
| Ramachandran[c] | | | |
| $\quad$ Favored (%) | 98 | 97 | 98 |
| $\quad$ Outliers (%) | 0.0 | 0.0 | 0.0 |
| Clashscore | 8.02 | 7.10 | 3.67 |
| Average B-factor | 76.23 | 129.10 | 46.02 |

Statistics for the highest-resolution shell are shown in parentheses
[a]$R_{sym} = (\Sigma|I_{hkl} - \langle I \rangle|)/(\Sigma I_{hkl})$, where the average intensity $\langle I \rangle$ is taken over all symmetry equivalent measurements, and $I_{hkl}$ is the measured intensity for any given reflection
[b]$R_{cryst} = (\Sigma||F_o|-k|F_c||)/(\Sigma|F_o|)$. $R_{free} = R_{cryst}$ for a randomly selected subset (5%) of the data that were not used for minimization of the crystallographic residual
[c]Categories were defined by PHENIX[66]. All non-glycine residues are included for this analysis

However, if the G-segment DNA is cleaved as intended, then the I-labels should appear on the +1, +7, and +12 locations in one of the complementary strands in each half of the DNA-gate (Fig. 2b). This expectation is validated by our observation that the difference peaks in the previously solved closed conformation of the Top2 cleavage complex are nicely overlaid with C5-methyl groups of the three thymidine residues (Fig. 2e). Moving now to our open structure, the difference peaks are again observed at the +7 and +12 locations. These signals, and the structural features of the bound G-segment DNA, perfectly match those seen in the closed form (Fig. 2e, f). Thus, both the SDS-PAGE analysis and the added structural data indicate that the new crystal form is indeed composed of the Top2 cleavage complex, and that the DNA fragment observed in the open conformation corresponds to the anticipated cleavage product of the enzyme. Similar to the structures commonly observed for the Top2 cleavage complexes with closed DNA-gate (Supplementary Figure 3d), a divalent metal ion, coordinated by the DxD di-acidic metal ion-binding motif of the TOPRIM domain near the non-scissile phosphodiester between the −1 and −2 nucleotides, was observed at the so-called "B-site"[23,28,29] in our structure (Supplementary Figure 3f). The presence of this B-site bound metal ion also agrees with the formation of Top2 cleavage complex.

Interestingly, the absence of detectable difference peaks above 2.0 σ in the vicinity of the catalytic tyrosine, between the iodinated and native structures of the open form, suggests that the three nucleotides located closest to the phosphotyrosyl bonds (+1 to +3) become highly disordered upon opening of the DNA-gate. Indeed, even in the closed conformation of the Top2 complex, nucleotides on the cohesive ends, despite remaining base-paired, have higher B-factor values than the double-stranded part enclosed in the G-segment binding groove[30]. This probably results from the paucity of DNA-protein interactions involving residues in the DNA-gate. Enhanced flexibility of the cohesive-end nucleotides may be functionally relevant, offering easier access of Top2 to the physically "open" state by reducing the free energy cost for disrupting the base-pair interactions and allowing the passage of the T-segment with less steric repulsion from the cohesive ends.

**Structural changes associated with the opening of DNA-gate.** Having established that the current hTop2β$^{core}$-DNA structure represents the Top2 cleavage complex in an open DNA-gate conformation, we proceeded to analyze the tertiary and quaternary differences between the closed and open states. Based on multiple, distinct conformational states observed for the catalytic core of Top2 in its apo (DNA-free) state[6,31], the opening of the DNA-gate has been predicted to follow a "book-opening" motion, in which the two halves of the DNA-gate simply move apart along the helical axis of G-segment[2,3]. According to this scheme, the cleaved G-segment fragments would remain co-linear, sharing the same helical axis during the opening of the DNA-gate. However, rather than remaining coaxial, the helical axes of the two DNA fragments in the open form reported here are found to be offset in opposite directions by ~20°, relative to the original G-segment axis (Fig. 1d). Thus, the observed open conformation results in part from sliding/shearing of the two WHD domains (Fig. 3a). Compared to the fully closed DNA-gate, the two A′α3 helices slide against each other by more than two helical turns, leading to solvent exposure of ~1800 Å$^2$ of what had been buried surface area within the closed DNA-gate, and complete disruption of the interface formed between the WHD domains.

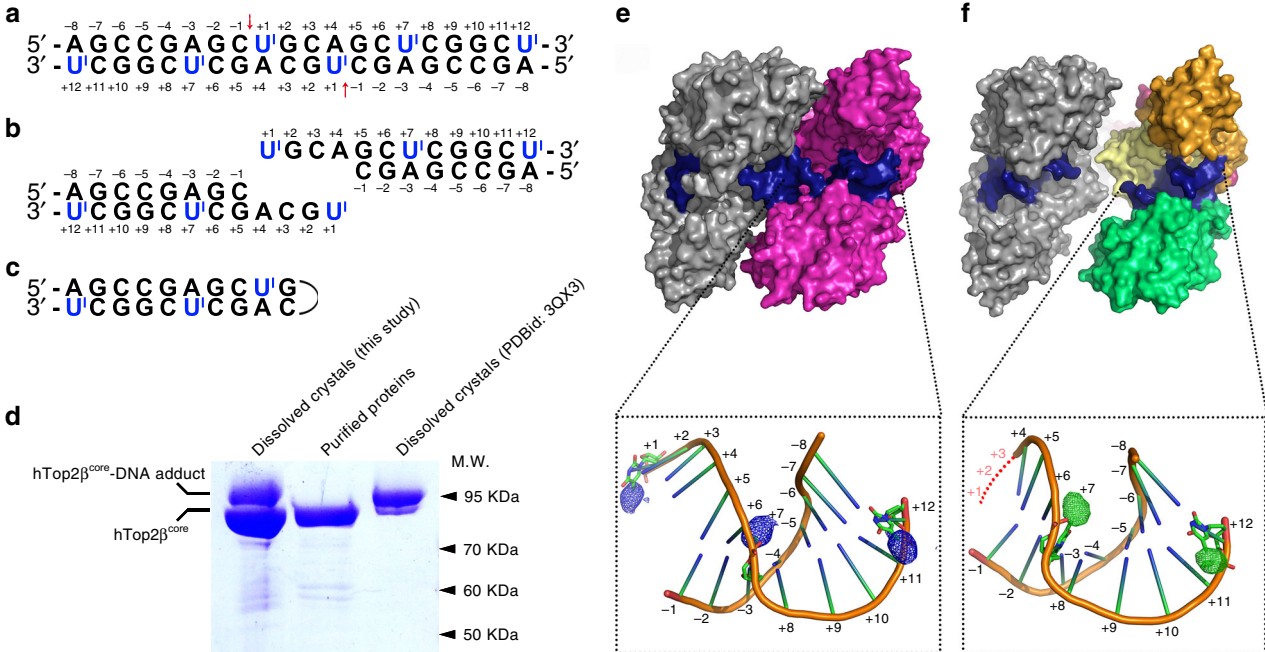

**Fig. 2** The reported structure represents a bona fide Top2 cleavage complex. **a** The sequence of iodine-labeled duplex DNA was derived from Fig. 1b by replacing thymidine nucleotides with 5-iododeoxyuridines (abbreviated as U[I]). Red arrows indicate the anticipated cleavage sites. **b** The cleaved fragments expected from Top2-mediated cutting of the iodine-labeled duplex DNA. The 5-iododeoxyuridines (U[I]) should be found at the +1, +7, and +12 positions. **c** Due to the palindromic nature of the DNA sequence, the oligonucleotide might form a hairpin with the three iodine-labeled nucleotides located at positions distinct from those in **b**. **d** SDS-PAGE analysis of the dissolved hTop2β[core]-DNA crystals. The presence of a species that migrates slower than the apo (DNA-free) Top2 but similar to the Top2-DNA adduct suggests that the newly obtained crystal is composed of the Top2 cleavage complex. The uncropped image has been provided as Supplementary Figure 10. **e** Surface representation of the etoposide-stabilized Top2 cleavage complex with a selected region of the bound DNA overlaid on the difference electron density map calculated by subtracting the native diffraction amplitudes from iododeoxyuridine-derivatized diffraction amplitudes ($F^{iododeoxyuridine}$–$F^{native}$) and phased with native phases (blue mesh, contoured at 4.0 σ). Difference peaks corresponding to the iodine sites can be found at the expected positions (C5 positions of the +1, +7, and +12 nucleotides). **f** Surface representation of the newly determined hTop2β[core]-DNA binary complex structure with a selected region of the bound DNA overlaid on the difference electron density map calculated by subtracting the native diffraction amplitudes from iododeoxyuridine-derivatized diffraction amplitudes ($F^{iododeoxyuridine}$–$F^{native}$) and phased with native phases (green mesh, contoured at 4.0 σ). Difference peaks corresponding to the iodine sites were found at the +7 and +12 nucleotides, indicating that the observed DNA fragment was produced by Top2-mediated cleavage, as shown in **b**. The lack of a difference peak at the +1 position provides strong support that the +1 ~ +3 nucleotides of the cohesive ends (red dashed line) are completely disordered in this structure

The space created between the two subunits due to the quaternary structural transition is augmented by tertiary changes within each subunit. When the C-gate and its flanking coiled-coil region of the closed and open structures are superimposed, an outward flexing of the G-segment-binding module about the hinge formed at the junctions of helices A′α14, A′α18, and A′α19 can be recognized (Fig. 3a). Thus, the sliding of the two subunits against each other, plus the swinging of the two halves of DNA-gate regions away from the structural dyad of Top2, both contribute to the opening of the DNA-gate.

**Structural features of the open DNA-gate**. The aforementioned structural changes in the DNA-gate region open a channel through which the T-segment DNA could pass, as part of the anticipated mechanism for resolution of DNA supercoiling. This presumed T-segment-conducting path is funnel-shaped, with a width shrinking from ~30 Å at its upper, or N-gate-facing portion (see Supplementary Figure 5 for the disposition of the N- and C-gate) to ~8 Å at the bottom (or C-gate-facing) side (Fig. 4a). A segment of B-form DNA can be introduced with no steric hindrance into the upper portion of the channel between the two cleaved G-segment fragments (Fig. 4b), giving rise to a model for entry of the T-segment into the DNA-gate. However, as the channel's bottom opening is narrower than the diameter of duplex DNA, further widening of this "bottleneck" region must

take place for T-segment to go through. Therefore, we speculate that this structure may correspond to an intermediate conformation occurring early during the passage of T-segment through the DNA-gate.

The lining of the channel shows no obvious preference in residue type (Supplementary Figure 4a, b), in sharp contrast with the G-segment binding groove, which is composed predominantly of basic residues. The absence of positively charged surface patches within the channel, together with the lack of discernable surface shape complementarity to B-form DNA, suggest at most a weak association between the channel and the T-segment, consistent with the expectation that T-segment should not be retained or slowed as it passes through the DNA-gate. However, compared to other parts of the molecular surface, a pronounced enrichment of conserved surface residues is seen on the channel-facing sides of the TOPRIM and WHD domains (Supplementary Figure 4c), indicating that the corresponding molecular surfaces are functionally important. Those clustered around the DxD di-acidic metal-binding motif may play supporting roles in G-segment binding and cleavage[32], while the ones decorating the surface of WHD may be crucial for stabilizing the DNA-gate in its closed conformation by mediating the dimerization of two WHD domains[6], such that the cleaved G-segment can be faithfully rejoined.

In addition to the funnel-shaped architecture, two structural characteristics of the T-segment conducting channel may bear

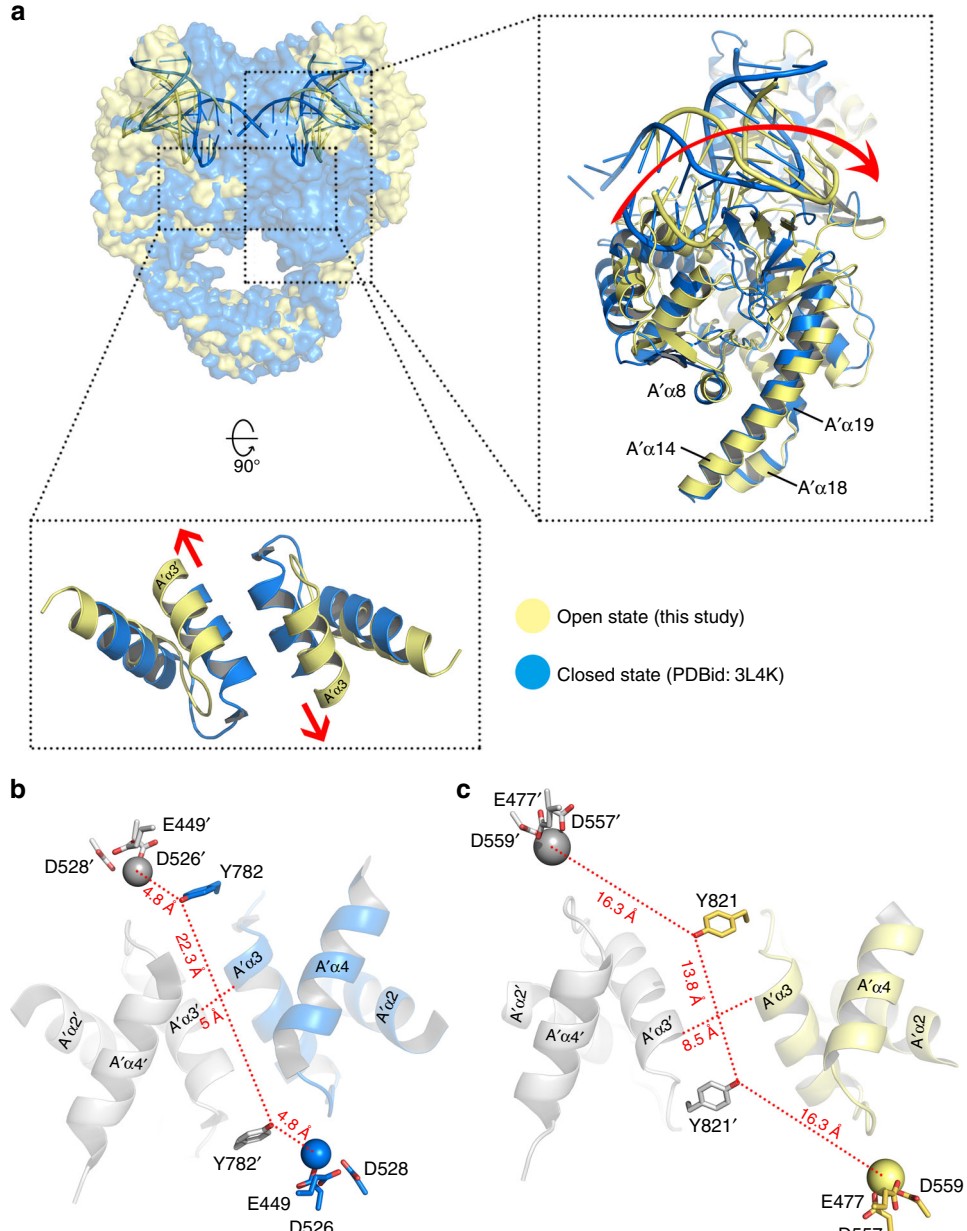

**Fig. 3** Quaternary and tertiary changes associated with DNA-gate opening. **a** Superposition of the eukaryotic C-terminal dimerization domain of the closed[12] (PDBid: 3L4K) and open (PDBid: 5ZEN, this study) conformations reveals that opening of the DNA-gate is achieved by the two subunits sliding against each other (enclosed view, bottom) plus an intra-subunit flexing (enclosed view, right). The closed and open DNA-gate conformations are colored blue and light yellow, respectively. The directions of quaternary sliding about the molecular dyad of Top2 and the tertiary flexing about the hinge located close to the junction of A'α14, A'α18, and A'α19 are indicated by arrows. **b, c** Top view (down the molecular dyad of the Top2 homodimer) of selected structural elements lining the bottom side of the DNA-conducting path and key catalytic groups in the closed (**b**) and open (**c**) conformations. For each conformation, the catalytic tyrosines, the divalent cations (shown as spheres), and the distance between two catalytic tyrosines are shown to illustrate structural changed in the DNA-gate during the closed-to-open transition. Residues from different monomers are colored differently, with labels belonging to the second monomer flagged by a prime

functional implications. First, rather than crossing the G-segment at right angles, the orientation of the channel suggests that the T-segment will cross the G-segment at an angle of about 60° upon its entry into the DNA-gate (Fig. 4c). This particular spatial disposition of T- and G-segment resembles a left-handed DNA braid (the L-braid) that is most frequently adopted by (+) supercoiled DNA molecules[33], and thus may help explain in part why Top2 relaxes (+) DNA supercoils more efficiently[34–36]. Given that the formation of the L-braid is energetically more favorable than the R-braid[37], the T-segment may spontaneously dock onto the G-segment with a L-braid-like crossing geometry even before the DNA-conducting channel is formed, and the T-segment can instantaneously enter the channel without the need to reorient as soon as the DNA-gate opens. On the other hand, the opening of the DNA-gate's bottom exit is essentially perpendicular to the G-segment (Fig. 4a), which suggests that the T-segment may undergo a counterclockwise rotation, relative to the enzyme, as it moves through the channel.

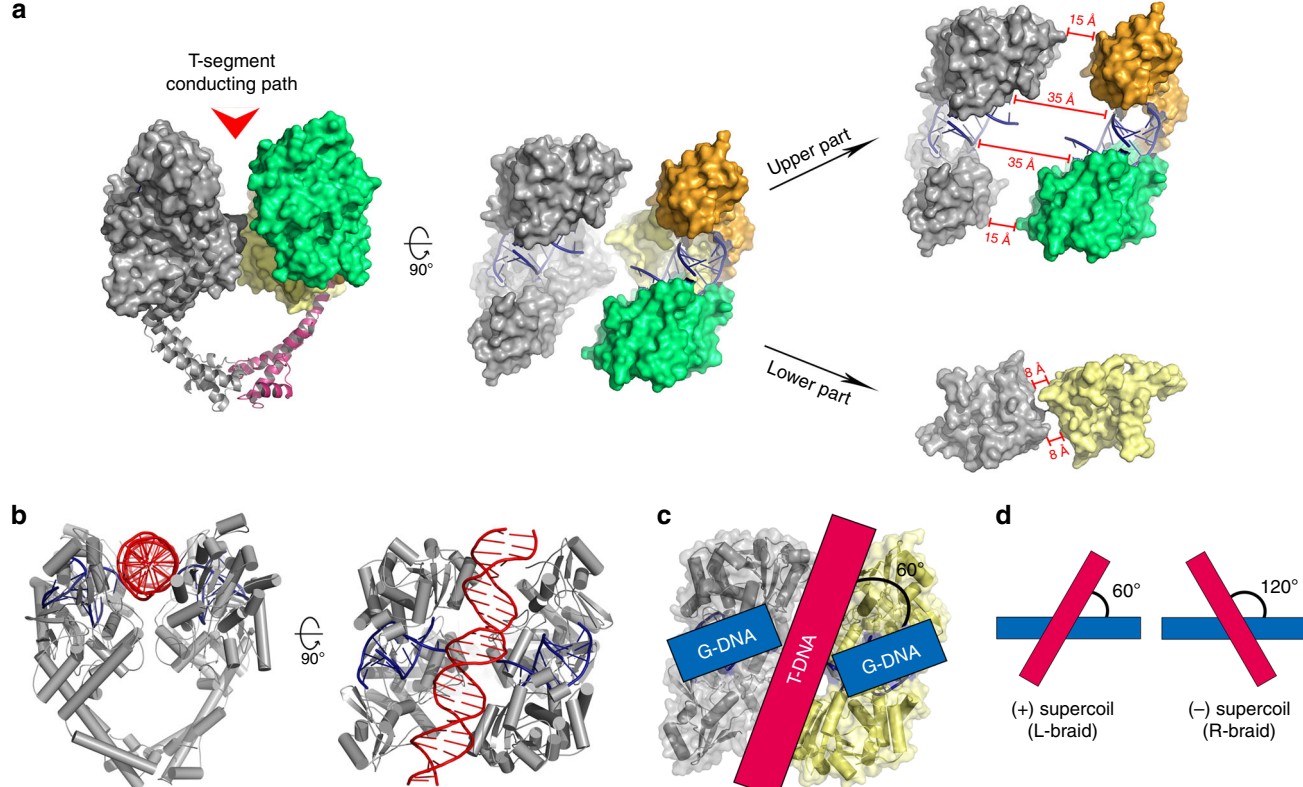

**Fig. 4** Structural features of the T-segment-conducting path. **a** Based on the differences in width and shape, the T-segment-conducting path may be divided into two parts: an upper and wider part and a lower and narrower part. Compared to the upper part of the DNA-gate, the T-segment-conducting path defined by the lower part inclines differently with respect to the G-segment, suggesting that the T-segment may undergo a counterclockwise rotation about the molecular dyad of Top2 as it moves across the DNA-gate. **b** A piece of B-form DNA can be introduced into the upper part of the T-segment conducting channel with no steric hindrance, giving a model that mimics the initial entry of the T-segment into the DNA-gate. **c** The ~60° inclination of the T-segment conducting channel about the G-segment resembles the left-handed crossover geometry often seen in positively supercoiled DNA (**d**). The two halves of the DNA-gate are colored differently according to the scheme used in Fig. 1

**Molecular dynamics of T-segment passage through the DNA-gate.** The structure reported here reveals opening of an entry to the DNA-gate, which allows docking another B-form DNA duplex in the space between the cleaved halves of G-segment, as if the T-segment is nicely poised to pass through the DNA-gate (Fig. 4b). This model of the ternary complex, comprising Top2, G-segment, and T-segment DNA, provides a reasonable starting point to simulate the passage of T-segment all the way through the gate. We used steered molecular dynamics (SMD)[38,39] to gain insight into how this passage occurs. Given the observation that the two TOPRIM domains in the current structure are further apart than in the structures showing a closed DNA-gate, and that they become considerably closer to each other in the structure showing the post-strand passage conformation of Top2[19], we speculate that a movement of TOPRIM domains toward each other coincides with the passage of the T-segment through the DNA-gate. This view appears to gain support from the finding of a domain-swapping event in the catalytic cycle of Top2[19], such that the structural transition from a passage-allowing intermediate into the domain-swapped, post-strand passage configuration is expected to draw the TOPRIM domains closer, through a gyrating motion of the N-gate region. Accordingly, the simulations simultaneously steered the T-segment through the open DNA-gate, while also drawing the TOPRIM domains closer, together, toward their relative positions in the post-strand passage state, (Fig. 5, Supplementary Figure 5 and Supplementary Movie 1).

The simulations show smooth evolution of structure and energy (Supplementary Figures 7 and 8a) as the T-segment moves through the channel, and three major stages of this process may be defined, based on the location of T-segment in the channel (Fig. 5, Supplementary Figure 6). Initially, the DNA-gate adopts an open, N-gate facing (ON) conformation, which allows initial entry of the T-segment (Fig. 5, step 1). No obvious structural change of the cleavage complex is seen in this state, as expected, because the opening of N-gate facing side is spacious enough for the T-segment to fit in without causing steric conflicts.

As the T-segment moves through the channel past the plane containing the cleaved G-segment, the two WHD domains separate, leading the middle and bottom portions of DNA-gate to widen (Fig. 5, step 2). However, the upper part of the channel does not widen appreciably. The concurrent closing movement of TOPRIM domains and the opening of WHD domains, exerting reciprocal effects on shape change of the whole DNA-gate, then bring the gate to a geometrically "occluded" (Occ) state. Here, the T-segment sits halfway through the channel, and the N-gate and C-gate facing ends of the channel are both only partly open. Concurrently, the cavity enclosed by the DNA-gate and the C-gate, which we call the central cavity, becomes widened and flatted, much as seen in a previously reported crystal structure of Bacillus gyrase[40] (Supplementary Figure 8b); this similarity supports the plausibility of the present simulations.

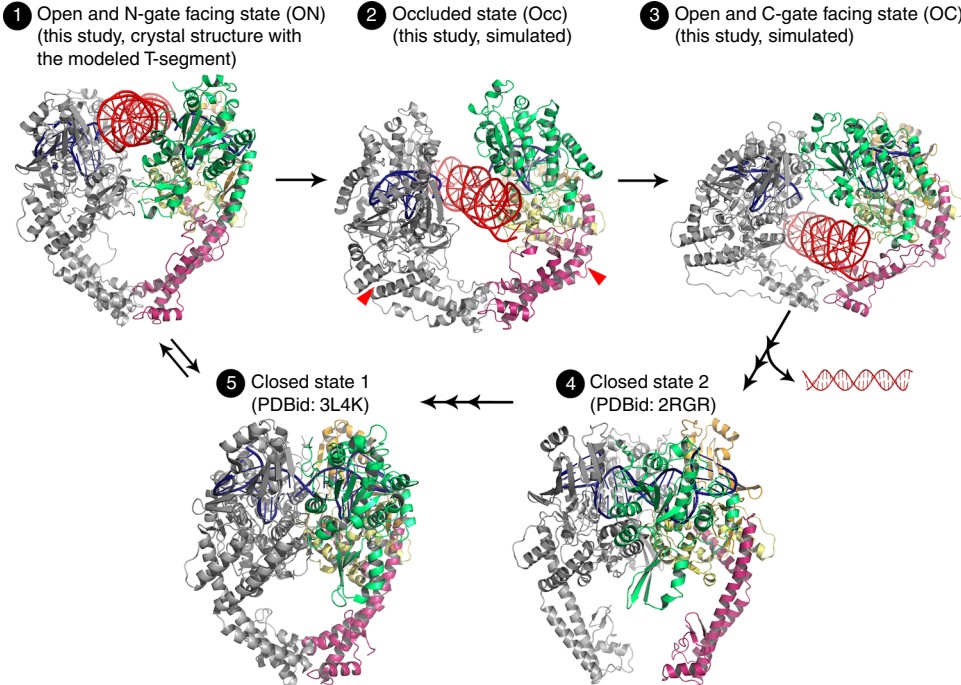

**Fig. 5** The DNA-gate may undergo a "rocker-switch"-type movement during T-segment passage. Starting from the newly determined structure, referred to as the open and N-gate-facing state (ON state, step 1), the full process of T-segment passage through the DNA-gate can be simulated by steered MD calculations. In the absence of a T-segment, we speculate that the DNA-gate would switch between the closed state (step 5) and ON state (step 1, a conformation ready for the T-segment to enter). In the presence of a T-segment and a force that drives the T-segment toward the C-gate, the DNA-gate would transform to an apparently occluded state (Occ state, step 2) and then an open and C-gate-facing state (OC state). Although subsequent structural transitions were not explored in our simulation, a post-strand passage conformation with an intact G-segment and opened C-gate[8] suggests that once the OC state is reached, the C-gate will open to allow the release of the T-segment and the DNA-gate will resume the closed conformation to reseal the cleaved G-segment (step 4). Finally, the closure of the C-gate (step 5) will reset the DNA-gate for the next strand passage event by reestablishing the equilibrium between the closed and ON states

With further progression of the T-segment into the central cavity (i.e., toward the C-gate), the upper portion of the channel continues to narrow, ultimately closing the N-facing entry to the DNA-gate, while the C-facing exit of the DNA-gate continues to widen, resulting in an open and C-gate facing (OC) conformation (Fig. 5, step 3). The T-segment then exits the DNA-gate and enters the central cavity. While the quaternary structure of the OC state appears quite different from the ON state, the conformation of each half-DNA gate in the former changes little relative to the latter, as shown by the RMSD plots (Supplementary Figure 8a). This indicates we are seeing mainly a rearrangement of the two protomers' relative position and orientation. The conformational changes of the DNA-gate in the course of this process are reminiscent of a "rocker-switch" mechanism employed by the ABC transporters, which transfer cargo molecules across the plasma membrane by cycling through an outward-facing, an occluded, and an inward-facing conformation[41–43].

## Discussion

The Top2-catalyzed double-stranded DNA passage reaction relies on the formation and opening of a protein-linked DNA-gate, which opens a physical gap in the G-segment wide enough for the T-segment to go through. This process comprises two main stages (Fig. 1a). First, Top2 unlocks the phosphodiester backbones of the G-segment by cleaving both DNA strands, resulting in the formation of DNA-gate in its closed conformation. The DNA-gate then undergoes a closed-to-open conformational change in which the two halves of cleaved G-segment separate to permit the entry and passage of the T-segment. Whereas plentiful structural

information has been obtained by X-ray crystallography and other techniques for the closed-conformation of DNA-gate[8,11,12,19,23,44,45], the atomic details regarding its opening have remained obscure. In this work, we report the visualization of the opening of Top2 DNA-gate (Fig. 1).

This conformational state of Top2 reveals several functionally significant structural features associated with the gate-opening event. First, the two cohesive ends of the cleaved G-segment, which are base-paired with each other in the closed state, become exceedingly mobile (Fig. 2f). Such an order-to-disorder transition may lower the steric barrier encountered by the T-segment during its transportation by reducing the repulsive forces between the cohesive ends and the T-segment as it enters the gate. Upon completion of strand passage, the DNA-gate must return to the closed conformation as a prerequisite for subsequent resealing of the cleaved G-segment. Here, the disordering might allow the two complementary cohesive ends to approach and pair with each other more efficiently and thus facilitate religation. Energetically, the disordering causes an increase in conformational entropy that is expected to favor the opening of the DNA-gate. Second, structural superimposition shows clearly that the observed gate-opening involves a quaternary sliding of the two halves of the DNA-gate against each other, plus a tertiary flexing of the DNA-gate region about a pivot located in the coiled-coil linker, rather than following the intuitively more straightforward open-book mode (see Supplementary Movie 4 in reference[16] as an example) (Fig. 3a). This unexpected conformational trajectory reshapes our understanding of how the opening of the DNA-gate is achieved. Third, the T-segment channel in the current structure is funnel-

shaped, such that width at its C-gate-facing side is narrower than the diameter of B-DNA. This indicates that further opening of the bottom region is necessary to complete the duplex passage reaction (Fig. 4a). Therefore, the full gating process would require sequential conformational changes of the DNA-gate. While opening of the N-gate-facing side does not rely on the presence of the T-segment, as demonstrated in the current structure, widening of the bottom region may be coupled to the transport of T-segment. Finally, since a DNA duplex can be reliably modeled into this structure between the cleaved ends of the G-segment, the full T-segment gating process can be simulated (Fig. 5).

Although, the opening of the Top2 DNA-gate has not previously been characterized at the atomic level, the conformational dynamics of the eukaryotic Top2 cleavage complex has been probed by single-molecule fluorescence resonance energy transfer (smFRET) analysis using FRET-pair-labeled double-stranded DNA as the G-segment[46,47]. The donor and acceptor fluorophores were placed 16-bp apart and symmetrically about a preferred Top2 cleavage site, an arrangement which allows direct monitoring of the opening of DNA-gate. The closed DNA-gate should exhibit a higher FRET efficiency due to the shorter distance between the FRET pair. Upon gate-opening, the two fluorophores must move away from each other, causing a decrease in FRET efficiency. The detection of reversible switching between the high and low FRET states, particularly in the absence of ATP and the T-segment, is consistent with our observation that a cleavage complex derived solely from the Top2 catalytic core and G-segment can be crystallized in both the closed and open conformations. The estimated distance between the FRET pair lengthens from ~53 Å in the closed conformation[12] to at least 66 Å in our newly determined structure (Fig. 1d). This change in distance is consistent with the observed change in FRET efficiency and suggests that the closed and open structures obtained by X-ray crystallography likely correspond to the high and low FRET state present in solution, respectively. The conformational change of the DNA-gate created by bacterial gyrase has also been monitored by smFRET using a similar FRET-pair-labeled DNA duplex[48]. The emergence of a DNA cleavage-dependent low FRET state was interpreted in terms of an intermediate conformation of the DNA-gate, in which the G-segment was thought to be significantly distorted, and which arises midway during the closed-to-open transition. We suspect that this intermediate species may also resemble the ON state, assuming the two low FRET states observed in eukaryotic Top2 and bacterial gyrase reflect the same molecular event. Taken together, both smFRET and crystallographic studies indicate the DNA-gate may enter the ON state spontaneously as long as the backbones of the G-segment are unlocked.

This study also presents the simulation of the passage of the T-segment through the DNA-gate. Steered MD simulations have been used to study Top2-binding drugs[49], but prior efforts to investigate the strand passage process through simulations, if any, may have been frustrated by insufficient information about the binding mode of T-segment to an 'opening' DNA-gate. This piece of the puzzle is provided by the current structure, which reveals a funnel-shaped channel ready for the entry of T-segment and thus makes it possible to construct a starting model with a precisely defined binding mode (Fig. 4b). Our simulations reveal that the DNA gate does not need to be fully open at any time for the T-segment to pass. Instead, we observe a rocker-switch motion of the DNA-gate, in which contact is always maintained across the passage, whether by direct protein–protein contacts, in the ON and OC states, or by protein-DNA-protein contacts, in the Occ state. One may conjecture that these motions help keep the Top2 dimer intact as the T-segment passes. We note, however, that there are still fundamental unanswered questions as to how the

hydrolysis of ATP drives the overall process, and that answering these questions will likely require additional structures, with the N-gate in different conformational states.

Comparing the conformations of the ON and the OC states, one may notice that the coiled-coil region underwent a helix-to-coil transition in the latter. We are not aware of experimental evidence for or against the unwinding of these helices during the catalytic cycle of Top2. However, there are precedents for functionally relevant local helix unwinding events. For example, the central bridge helix of RNA polymerase I can adopt either a fully folded conformation, in the elongation complex, or an unwound state, in the free enzyme[50].

The DNA cleavage activity of Top2 is usually referred to as a double-edged sword for living organisms[51]. While this activity is required for the formation of the DNA-gate and thus is indispensable for Top2 function, it may cause DNA lesions if the DSB embedded within the DNA-gate is not resealed properly. Since the cleavage and religation reactions are executed by the same enzyme active site, an interesting dilemma has been raised regarding how efficient G-segment cleavage and strand passage by Top2 can be achieved without jeopardizing fidelity in resealing the cleaved G-segment. In one proposal, the free energy derived from ATP binding and hydrolysis may be utilized to maintain the overall dimeric architecture of Top2, so that the risk of having the two halves of cleaved G-segment becoming permanently separated due to subunit dissociation is greatly reduced[52]. Still, the ultimate prevention for DNA double-strand break relies on whether DNA-gate may resume the closed conformation after strand passage to allow G-segment religation. In this regard, we speculate that the equilibrium between the closed and ON states and the simulated rocker-switch movement of the DNA-gate may contribute to the balance between catalytic efficiency and safety. With its DNA-conducting channel being half-opened and ready for T-segment to enter, the ON state is primed for engaging in strand passage. If, however, the T-segment has not yet been captured, the two A′α3 helices that line the narrower part of the DNA-conducting channel are in place to re-associate, which would restore the interface between the two WHD domains and return the DNA-gate to its closed, religation-competent conformation. In addition, according to our simulation, advancing the DNA-gate toward the Occ and OC conformation depends on the presence of not only the T-segment, but also a force that drives the T-segment toward the C-gate. Therefore, the DNA-gate would access the energetically more strained states only when necessary. Once T-segment passes through the DNA-gate, it is desirable to rejoin the cleaved G-segment without much delay. The OC conformation appears well suited for this role because the N-gate-facing side of the DNA-gate has already closed, which puts the two cohesive ends in proximity for religation. In sum, we suggest the simulated rocker-switch movement may add a layer of safety in suppressing Top2-induced DNA double-strand breakage.

The Top2-catalyzed DNA cleavage and religation require the presence of divalent metal ions[53,54]. Two-metal ion-binding sites, designated as A-site and B-site, have been recognized in the crystal structures of Top2-DNA complexes[8,11,12,20,22,23,28,29] (Supplementary Figure 3). The metal ion bound in the A-site is coordinated by the scissile phosphate and the N-terminal Asp residue of the DxD motif. The B-site is located near the non-scissile phosphodiester between the −1 and −2 nucleotides, where the bound metal ion is coordinated by both Asp residues of the DxD motif. It is widely accepted that the A-site metal ion is essential for G-segment cleavage and religation by stabilizing the penta-coordinate transition state[11,12,22]. In contrast, the functional significance for having a metal ion present in the B-site during catalysis is more controversial; one speculation is that its

presence may help anchoring the substrate DNA to enhance cleavage efficiency[12]. Whether Top2 requires simultaneous binding of two-metal ions or a single metal ion that shuffles between the two sites has remained unsettled. Nevertheless, the existence of a metal ion in the A-site is observed only when Top2 assumes the pre-cleavage or presumably the immediate post-cleavage state[11,12,22], in which the scissile phosphate and DxD motif are optimally aligned for metal ion coordination (Supplementary Figure 3b, c). Following G-segment cleavage, the +1 and −1 nucleotides are free to move away from each other. In the structure commonly adopted by the Top2 cleavage complexes[8,11,20,23,28,29] (Supplementary Figure 3d, e), the integrity of the A-site is disrupted due to a wider separation between the +1 and −1 nucleotides, in these cases the metal ion is seen exclusively in the B-site. The constant presence of a metal ion in the B-site in these closed-form structures has also been inferred from molecular simulations, including studies using picosecond-scale QM calculations[55] and microsecond-scale classical MD simulations[56]. The finding that the B-site is occupied by a metal ion in our structure (Supplementary Figure 3f), and that the metal ion remained in the B-site throughout our sub-microsecond steered MD simulations of the DNA passage process (Supplementary Figure 7) suggest that the B-site metal ion is likely held tightly by Top2 during the opening of the DNA-gate. It has been proposed that the recruitment of a metal ion from B-site to the A-site would be sufficient for driving G-segment religation[57]. MD simulation further implicates that the metal ion bound in the B-site may translocate to the A-site[55]. It is thus conceivable that holding the B-site metal ion in place during T-segment passage may facilitate subsequent G-segment religation: the metal ion shuffling can take place as soon as the A-site is reestablished upon the closure of the DNA-gate. The issue regarding metal ion occupancy in Top2 catalytic cycle, however, would require further investigation because the time scales of these simulations were short compared to the natural strand passage process.

The Top2s are validated cellular targets of many clinically active anticancer drugs and antibiotics[24,25]. Crystallographic studies have revealed that these drugs interact with the closed conformation of the DNA-gate and inhibit the religation of cleaved G-segment by inserting into the DNA cleavage sites[11,21,23]. The accumulation of drug-induced DNA double-strand breaks in turn leads to cell death. Despite the proven effectiveness of these Top2-targeting drugs, their long-term efficacy and broader application have been compromised by the emergence of drug-resistant mutations[58,59]. Therefore, it should be clinically valuable to develop new strategies for stabilizing the Top2-mediated DNA breaks, and trapping the ON conformation may represent a promising approach. The feasibility of this scenario may be inferred from the poisoning effects of the peptide antibiotic microcin B17 and CcdB toxin on bacterial gyrase[60,61]. While it has remained unknown how microcin B17 induces gyrase-mediated DNA breaks, this compound likely "recognizes a conformation of the enzyme that is only revealed during the strand-passage process, via a transiently exposed hydrophobic regions of the enzyme"[61,62]. Mutagenesis studies further indicate the association is achieved via the TOPRIM domain of gyrase. The T-segment binding groove revealed by the ON conformation seems like a reasonable candidate site for microcin B17 binding. The crystal structure of CcdB in complex with the gyrase C-gate region suggests that the toxin stabilizes the cleavage complex by binding to the bottom cavity of gyrase[63]. Intriguingly, we found that the CcdB-bound C-gate region can be superimposed exceptionally well onto the ON conformation, and that CcdB can be fitted snugly inside the cavity (Supplementary Figure 9). Moreover, CcdB appears to target exclusively the strand

passage-competent conformation of gyrase and does not interact with the apo enzyme[64]. Taken together, we suspect that CcdB may exert its bactericidal activity by trapping the ON conformation. In conclusion, this study not only provides insights into the structural basis of Top2-catalyzed DNA strand passage, but may also contribute to the development of new and medically relevant Top2-targeting agents.

## Methods

**Sample preparation.** For preparing recombinant hTop2β$^{core}$, E. coli BL21 (DE3) Star-pLysS cells harboring the 51bDBCCβ plasmid[23], which contains the coding sequence of hTop2β$^{core}$ (residues 445–1201, Fig. 1b), were grown in LB medium at 37 °C until OD$_{600}$ reached 0.5. Protein expression was induced by adding isopropyl β-D-1-thiogalactopyranoside (IPTG) to 0.3 mM. The cell culture was shifted to 20 °C for 16 h for accumulating the recombinant protein. Cells were collected by centrifugation and then re-suspended in lysis buffer (50 mM sodium phosphate, pH 7.4, 10% glycerol, 500 mM NaCl, 5 mM β-mercaptoethanol, 0.5 mM PMSF, and 10 mM imidazole). After disrupting the cells by sonication, the crude lysate was centrifuged at 18,000 rpm for 2 h at 4 °C. The clarified lysate was applied to a Ni-NTA column. The column was washed to baseline with wash buffer (lysis buffer containing no PMSF), and the bound protein was eluted with elution buffer (wash buffer containing 250 mM imidazole). Fractions containing hTop2β$^{core}$ and were pooled and dialyzed against buffer A (30 mM Tris-HCl, pH 7.5, 15 mM NaCl, 2 mM β-mercaptoethanol, and 1 mM EDTA) at 4 °C for 4 h, and then loaded onto a HiPrep 16/10 Heparin FF column. The protein was eluted in a linear gradient over 10 column volumes with buffer B (buffer A containing 1 M NaCl). The eluted protein was further purified by a size-exclusion column (Hi-Load Superdex 200). Fractions containing the functional, homodimeric form of hTop2β$^{core}$ (MW ~180 kDa) were collected and concentrated to ~8.5 mg/ml (in gel-filtration buffer: A buffer containing 70 mM NaCl) for subsequent applications.

The DNA oligonucleotides (native: 5′-AGCCGAGCTGCAGCTCGGCT-3′; 5-iododeoxyuridine(U$^I$)-labeled: 5′-AGCCGAGCU$^I$GCAGCU$^I$CGGCU$^I$-3′) were purchased from Integrated DNA technologies. The duplex DNA substrates used for crystallization were prepared by first dissolving each oligonucleotide in gel-filtration buffer, followed by sequential incubation at 94 °C (10 min), 50 °C (10 min) and 25 °C (5 min) before being stored at −20 °C.

**Crystallization of the hTop2β$^{core}$-DNA cleavage complex.** The protein solution for crystallization is composed of purified hTop2β$^{core}$ (5 mg/mL, ~28 μM), 2 mM etoposide or doxorubicin (diluted from 50 mM stock in pure DMSO), and ~34 μM of the 20-bp duplex DNA substrate (native or U$^I$-labeled) in the C buffer (30 mM Tris-HCl (pH 7.0), 70 mM NaCl, 2 mM MnCl$_2$, 1 mM EDTA, 2 mM β-mercaptoethanol). For obtaining crystals of the open conformation of the hTop2β$^{core}$-DNA cleavage complex, 1 μl of this protein solution was mixed with an equal amount of the reservoir solution (100 mM magnesium acetate, 50 mM 2-(N-morpholino)ethanesulfonic acid pH 5.8, and 22% 2-methyl-2,4-pentanediol) and equilibrated against 200 μl of the reservoir solution using the hanging-drop vapor diffusion method with VDX$^{TM}$ plate (Hampton Research) at 4 °C. Oval-shaped single crystals (space group $P3_221$) suitable for data collection usually appear within 2 weeks. The method for obtaining crystals of the closed conformation of the hTop2β$^{core}$-DNA cleavage complex (rod-shaped, space group $P2_1$) has been described in our previous works[23]. The two crystallization procedures are very similar except that a mild adjustment of the reservoir solution (100 mM magnesium acetate, 50 mM 2-(N-morpholino)ethanesulfonic acid pH 5.6, and 26% 2-methyl-2,4-pentanediol) and the use of MgCl$_2$ in place of MnCl$_2$ in the protein sample buffer appear to favor the formation of the $P2_1$ crystals. Both types of crystals were collected by transferring into a substitute mother liquor containing 20% 2-methyl-2,4-pentanediol and 20% glycerol before looping and flash-freezing in liquid nitrogen for data collection.

**X-ray crystallography.** All diffraction datasets were collected at NSRRC, Taiwan (beamlines BL13C1 and BL15A1) and processed using the HKL2000 program suite[65]. For the newly obtained $P3_221$ crystals, the asymmetric unit contains only half of the homodimeric hTop2β$^{core}$-DNA cleavage complex. Thus, the structure was solved by molecular replacement with phenix.automr[66] using one subunit of hTop2β$^{core}$ taken from the hTop2β$^{core}$-DNA-etoposide structure (PDBid: 3QX3)[23] as the search model followed by model building with phenix.autobuild. The MR-phased initial structure then underwent rounds of manual adjustment and refinement using Coot[67] and PHENIX[66]. Missing residues in the final $P3_221$ structure (PDBid: 5ZEN) are as follows: 445–451, 597–602, 616–624, 703–706, and 1112–1134 in chain A. The $P2_1$ structure was re-determined using the U$^I$-labeled duplex DNA substrate (PDBid: 5ZRF). This structure was solved by subjecting the previously determined hTop2β$^{core}$-DNA-etoposide structure (PDBid: 3QX3)[23] to rigid-body and positional refinement against the new data, and the resulting $Fo$–$Fc$ map shows ambiguously the positions of the 5-iodo groups. The $P3_221$ structure (PDBid: 5ZQF) was re-determined and analyzed using the same approach. The data collection and refinement parameters are listed in Table 1. All figures were generated using Pymol[68].

**Steered molecular dynamics simulation and structural analysis**. The starting model for simulating the passage of T-segment through DNA-gate was constructed using the current structure by placing a B-DNA segment between the two halves of cleaved G-segment (Fig. 5, step 1). Water molecules observed in the crystal structure were retained, and missing atoms were modeled back based on the conformation generated in a prior simulation of the Top2β-DNA complex[56]. Hydrogens were added to the structure according to the protonation state predicted using PDB2PQR[69,70] v2.0. The manganese ions in the crystal structure were substituted with magnesium ions to better mimic the cellular environment. Although prior work using QM/MM and classical MD simulations have suggested a two-metal ion cleavage mechanism[12,55], we modeled a single $Mg^{2+}$ ion at each of the two catalytic sites, based on the present crystal structure, as well as on prior structural studies of this and other topoisomerases in various states of the catalytic cycle[22,23,56]. These two $Mg^{2+}$ ions remained in their binding sites throughout the simulations reported here. The structural model of a 26-bp stretch of B-DNA, consisting of the sequence 5′-AGCATACACGTACCTTAGTACAGGGT-3′ in one of its double strands, was built using the Coot program[67] and was used as the T-segment.

Parameter files for molecular dynamics simulations were created using the LEaP program of the AMBER biomolecular simulation package[71,72], with the ff99SB force field for protein[73,74] and the parmbsc0 modifications for nucleic acids[75]. The parameters of the covalently bonded active-site Tyr821 and cleavage-site nucleotide were generated in a prior study[56]. The complex, comprising Top2, the G-segment, the T-segment, and the retained crystal waters and cations, was solvated with explicit TIP3P[76] waters in an octahedral box. Sodium and chloride ions were added to the system to simulate 0.15M sodium chloride, and counter ions were added to neutralize the charge of the solvated system. The whole system comprises 205,957 atoms.

Molecular dynamics (MD) simulations were carried out using the CUDA accelerated PMEMD program of the AMBER 16 package, with the particle-mesh Ewald method for calculating the full electrostatic interactions of a periodic box in the macroscopic lattice of repeating images. Before starting MD, energy minimization of the system was carried out using the steepest descent algorithm. Thermalization to 310 K was conducted through NVT simulation with the Berendsen thermostat. The system was then equilibrated at a pressure of 1 bar through NPT simulations with the Berendsen barostat. Steered MD simulations[38,39] of the strand passage process were conducted by introducing two virtual springs with a fixed force constant to control the movements of the tethered parts in the complex over time. One of the springs was placed between the centroids of the TOPRIM domains on each side of the complex to bring them closer; the other was used to bring the T-segment from top of the G-segment toward the base of the WHD domains. The distance between the TOPRIM domains was reduced at the same rate as the T-segment was brought through the DNA-gate. An additional extended run was performed after the T-segment had reached the base of the WHD domains. Here, the virtual spring used to tether the TOPRIM domains in the prior part of the simulation was removed, and the spring tethered to the T-segment was used to move it further away from the WHDs and into the cavity enclosed by the DNA-gate and C-gate. Positional restraints were applied to C5′ atoms of the T-segment to retain the B-form tertiary structure throughout the simulations. The hydrogen mass repartitioning method[77] and SHAKE constraints[78] was employed to enable simulations with a time step of 4 fs. The simulations were carried out for 160 ns. Atomic coordinates were saved to disk every 10 ps for subsequent analysis using the CPPTRAJ program[79]. Molecular graphics in Supplementary Movie 1 were generated using UCSF Chimera[80].

**Data availability**. Atomic coordinates and structure factors that support the findings of this study have been deposited in the Protein Data Bank (PDB) with the accession codes 5ZEN (hTop2βcore-DNA-binary complex), 5ZQF (hTop2βcore-DNAI-binary complex), and 5ZRF (hTop2βcore-DNAI-etoposide ternary complex). Other data are available from the corresponding authors upon reasonable request.

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

## Acknowledgements

Portions of this research were carried out at beamlines 15A1 and 13C1 of the National Synchrotron Radiation Research Center (Taiwan), beamline SP12B2 of the SPring-8 (Japan) and the San Diego Supercomputing Center (US). We are grateful to the staffs of Technology Commons in College of Life Science and Center for Systems Biology, National Taiwan University. We thank Prof. Hung-Wen Li, Dr. Wan-Chen Huang and members of our laboratories for helpful discussion. This work was supported by the Ministry of Science and Technology (106-2113-M-002-021-MY3 and 104-2911-I-002-302 to N.-L.C.), National Taiwan University (104R7614-3 and 104R7560-4 to N.-L.C.), Academia Sinica Postdoctoral Fellowship to N.-L.H., and NIH grant GM061300 to M.K.G. The contents of this paper are solely the responsibility of the authors and do not necessarily represent the official views of the NIH. M.K.G. has an equity interest in and is a cofounder and scientific advisor of VeraChem LLC. This paper is dedicated to the memory of Prof. Tao-shih Hsieh, a model scientist for us.

## Author contributions

S.-F.C. and N.-L.C. devised crystallographic and biochemical experiments. N.-L.H., J.-H.L., and M.K.G. devised the simulation strategies and performed steered molecular dynamics. C.-C.W. cloned plasmids and S.-F.C. and Y.-J.Y. expressed, purified, and crystallized the protein. S.-F.C., Y.-R.W., and N.-L.C. performed X-ray crystallography. S.-F.C., N.-L.H., M.K.G., and N.-L.C. analyzed the data and wrote the manuscript.

## Additional information

**Competing interests:** M.K.G. has an equity interest in and is a cofounder and scientific advisor of VeraChem LLC. The remaining authors declare no competing interests.

