## [Peer Review File · Nature Communications]

Reviewers' comments:

Reviewer #1 (Remarks to the Author):

Topoisomerases are key enzymes to control DNA topology, thus representing validated drug targets in humans and bacteria. The mechanism used by these enzymes to cleave and re-join the DNA is however complex and largely unclear, yet.

This study presents the first structure of the enzyme topoII in an open state of the DNA gate, which was never observed before. This new structure represents a starting point for further studies to gain a deeper understanding of TopoII mechanism. In addition, the authors make use of MD simulations to investigate and propose a 'rocker-switch' mechanism for T-segment passage through the DNA gate – a step of the catalytic cycle of topoII that is still widely uncharacterized. Overall, the integration of these new structural data on this open conformational state of topoII, with MD simulations, provides new insights into the dynamics of the T-segment for its passage through the DNA gate. This hypothetical mechanism should generate broad discussions in the field, stimulating new experiments and studies to prove (or disprove) this possible mechanism for T-segment passage.

The MD simulations are technically sound. Pictures are clear. The manuscript is well presented; it allows to capture the main message of the story. Reading the manuscript, however, I was expecting to find something about the presence and role of catalytic metal ion(s). Indeed, MD simulations (including steered MD for drug binding at the cleavage complex) have been used already in the past to look at the dynamics of topoII for function and inhibition (see e.g. Palermo et al in JCTC 2013, 857 and Chem. Comm., 2015, 51, 14310, which are missing from the ref section). I recognize that the simulations presented here look at the passage of the T-segment through the DNA-gate, which is highly challenging given the many degrees of freedom involved. Anyhow, I wonder if the authors can say something more – based on this new structure and simulations – on the still questioned presence of two (vs one) ions as in Schmidt et al (Ref 12 of the manuscript), and how the catalytic site appears in this newly-characterized open state in regards to the metal-aided structure of the catalytic site. From figure 3, I see one ion only, at each site. I understand that this open state occurs after DNA cleavage, and therefore corresponds to a more 'flexible' state where ions are likely to be highly movable. However, I invite the authors to comment on this aspect, which I think would enrich the overall story.

One more question regards the model shown at step 3, in figure 5. This is the result of the simulations of the passage of the T-segment, which leads to the structural model that is reported. Maybe it is because of the viewpoint shown in the figure, but this model seems quite distorted compared to the other ones in the same figure. The protein's shape seems somehow squeezed, with the T-segment included at the center. The authors say "the upper portion of the channel continues to narrow, ultimately closing the N-facing entry to the DNA-gate". Is this conformation of this upper portion any similar to the crystallographic one at step 1? I would appreciate the authors' comment on this particular conformation, and the way it is formed.

The reported structure is of the beta isoform of TopoII. The authors report (abstract and page 9, lines 245-256) that the new structure in part explains why topo2 preferentially relaxes positive supercoils. However, as shown by Osheroff and co-workers (Reference 32), the alpha isoform shows a preference for positive supercoils. On the other hand, it seems that the beta isoform (the structure reported in this paper) shows no such preference. Further, Timsit reported (Nucleic Acids Res. 2011 Nov; 39(20): 8665–8676) that for the initial clamping of the G- and T-segments, right-handed crosses are optimal. For the T-segment expulsion, repulsive interactions between the DNA segments were proposed in a left-handed cross-over. Any comments on this?

Minor Corrections:

1. Page 8, Line 212- There is a mention of A' α 19 in Fig 3a. However, the helix is not marked in the figure.

2. Reference 68 and 69 have to be superscripted
3. From the supplementary Figure 7, I understand that 160 ns long simulations are performed. It should be mentioned in the method section, too.

Reviewer #2 (Remarks to the Author):

This is a very interesting structural study in the topo II area as it is a first structure determination of an open G-gate. As the authors correctly state all the previous structures are for closed G-gates and hence do not address this question of how the T-segment DNA traverses the open G-gate although there has been based on molecular mechanics modelling a view of the trajectory and how this may occur in the 2nd movie in Laponogov et al. *Nucleic Acids Research*, Volume 41, Issue 21, 1 November 2013, Pages 9911–9923, <https://doi.org/10.1093/nar/gkt749>.

The X-ray crystallography has been very carefully executed with iodo-U substituted DNAs use to unambiguously determine the positioning of the DNA and hence the veracity of the structures. The refinement has also been very carefully performed and the structures are well refined.

I could see the crystallographic parameters for the P3sub2 structure in the supplementary, but not that for the P2sub1 structure talked about in the text but the crystallographic parameters are not tabulated. Also in the text the space group is referred to as P3 and not P3sub2! This should be corrected as P3 is also a possible space group (also the Laue group) but the systematic absences and MR provide the definitive space group as being P3sub2.

The steered molecular dynamics simulations are very interesting but I am very surprised that a pair of the helices forming the C-gate completely unwrap at the end of the simulation. I have run molecular dynamics on systems although not on these topo II-DNA and topo II-DNA-drug systems, but to me this would be intuitively unlikely as these are very long helices stabilised by many H bonds between the carbonyls and the main chain NHs all along the length of the helices. Are the force field constraints being correctly applied at these ending stages? Is this a true reflection of what happens in solution, this unwrapping on these time scales?

I think that this paper is a extremely important contribution to the field of topoisomerase structure and mechanism

We greatly appreciate the obvious time and effort that the Reviewers spent on our manuscript. And thanks to their keen attention and valuable comments, we believe our manuscript is further improved after revisions.

A detailed point-by-point discussion of the changes we made is given below. The text of the points raised by the Reviewers is in bold and enclosed within brackets, followed by our responses.

Response to Reviewer #1:

[The MD simulations are technically sound. Pictures are clear. The manuscript is well presented; it allows to capture the main message of the story.]

We thank the reviewer for this positive assessment.

[Reading the manuscript, however, I was expecting to find something about the presence and role of catalytic metal ion(s). Indeed, MD simulations (including steered MD for drug binding at the cleavage complex) have been used already in the past to look at the dynamics of topoII for function and inhibition (see e.g. Palermo et al in JCTC 2013, 857 and Chem. Comm., 2015, 51, 14310, which are missing from the ref section). I recognize

that the simulations presented here look at the passage of the T-segment through the DNA-gate, which is highly challenging given the many degrees of freedom involved. Anyhow, I wonder if the authors can say something more – based on this new structure and simulations – on the still questioned presence of two (vs one) ions as in Schmidt et al (Ref 12 of the manuscript), and how the catalytic site appears in this newly-characterized open state in regards to the metal-aided structure of the catalytic site. From figure 3, I see one ion only, at each site. I understand that this open state occurs after DNA cleavage, and therefore corresponds to a more “flexible” state where ions are likely to be highly movable. However, I invite the authors to comment on this aspect, which I think would enrich the overall story.]

We thank the reviewer for directing us to these highly relevant and insightful papers, which are now referenced in the revised manuscript (references #49 and #55).

And thank to the reviewer’s excellent suggestion, we have added a new figure (Supplementary Fig. 3) to show not only the metal ion binding site in our structure, but also how the arrangement of metal ion(s) may change as Top2 proceeds through different catalytic stages during the DNA cleavage reaction. In addition, we have added two new paragraphs (shown below) to discuss the potential roles of metal ions in Top2-catalyzed DNA cleavage and religation. We believe these additions make the whole story more complete.

Page 6, line 176 ~ page 7, line 182: “Similar to the structures commonly observed for the Top2 cleavage complexes with closed DNA-gate (Supplementary Fig. 3, panel d), a divalent metal ion, coordinated by the DxD diacidic metal ion-binding motif of the TOPRIM domain near the non-scissile phosphodiester between the -1 and -2 nucleotides, was observed at the so-called “B-site”^{23,28,29} in our structure (Supplementary Fig. 3, panel f). The presence of this B-site bound metal ion also agrees with the formation of Top2 cleavage complex.”

Page 16, line 446 ~ page 17, line 483: “The Top2-catalyzed DNA cleavage and religation require the presence of divalent metal ions^{53,54}. Two metal ion-binding sites, designated as A-site and B-site, have been recognized in the crystal structures of Top2-DNA complexes^{8,11,12,20,22,23,28,29} (Supplementary Fig. 3). The metal ion bound in the A-site is coordinated by the scissile phosphate and the N-terminal Asp residue of the DxD motif. The B-site is located near the non-scissile phosphodiester between the -1 and -2 nucleotides, where the bound metal ion is coordinated by both Asp residues of the DxD motif. It is widely accepted that the A-site metal ion is essential for G-segment cleavage and religation by stabilizing the penta-coordinate transition state^{11,12,22}. In contrast, the functional significance for having a metal ion present in the B-site during catalysis is more controversial; one speculation is that its presence may help anchoring the substrate DNA to enhance cleavage

efficiency¹². Whether Top2 requires simultaneous binding of two metal ions or a single metal ion that shuffles between the two sites has remained unsettled. Nevertheless, the existence of a metal ion in the A-site is observed only when Top2 assumes the pre-cleavage or presumably the immediate post-cleavage state^{11,12,22}, in which the scissile phosphate and DxD motif are optimally aligned for metal ion coordination (Supplementary Fig. 3, panels b and c). Following G-segment cleavage, the +1 and -1 nucleotides are free to move away from each other. In the structure commonly adopted by the Top2 cleavage complexes^{8,11,20,23,28,29} (Supplementary Fig. 3, panels d and e), the integrity of the A-site is disrupted due to a wider separation between the +1 and -1 nucleotides, in these cases the metal ion is seen exclusively in the B-site. The constant presence of a metal ion in the B-site in these closed-form structures has also been inferred from molecular simulations, including studies using picosecond-scale QM calculations⁵⁵ and microsecond-scale classical MD simulations⁵⁷. The finding that the B-site is occupied by a metal ion in our structure (Supplementary Fig. 3, panel f), and that the metal ion remained in the B-site throughout our sub-microsecond steered MD simulations of the DNA passage process (Supplementary Fig. 7) suggest that the B-site metal ion is likely held tightly by Top2 during the opening of the DNA-gate. It has been proposed that the recruitment of a metal ion from B-site to the A-site would be sufficient for driving G-segment religation⁵⁶. MD simulation further implicates that the metal ion bound in the B-site may translocate to the A-site⁵⁵. It is thus conceivable that holding the B-site metal ion in place during T-segment passage may facilitate subsequent G-segment religation: the metal ion shuffling can take place as soon as the A-site is reestablished upon the closure of the DNA-gate. The issue regarding metal ion occupancy in Top2 catalytic cycle, however, would require further investigation because the time scales of these simulations were short compared to the natural strand passage process.”

We have also added the following text in the Methods section to describe how the metal ions were treated during the simulation.

Page 21, lines 591 ~ 598: “The manganese ions in the crystal structure were substituted with magnesium ions to better mimic the cellular environment. Although prior work using QM/MM and classical MD simulations have suggested a two-metal ion cleavage mechanism^{12,55}, we modeled a single Mg²⁺ ion at each of the two catalytic sites, based on the present crystal structure, as well as on prior structural studies of this and other topoisomerases in various states of the catalytic cycle^{22,23,68}. These two Mg²⁺ ions remained in their binding sites throughout the simulations reported here.”

[One more question regards the model shown at step 3, in figure 5. This is the result of the simulations of the passage of the T-segment, which leads to the structural model that is reported. Maybe it is because of the viewpoint shown in the figure, but this model

seems quite distorted compared to the other ones in the same figure. The protein's shape seems somehow squeezed, with the T-segment included at the center. The authors say "the upper portion of the channel continues to narrow, ultimately closing the N-facing entry to the DNA-gate". Is this conformation of this upper portion any similar to the crystallographic one at step 1? I would appreciate the authors' comment on this particular conformation, and the way it is formed.]

The Reviewer's points are well taken. As the T-segment moves through the DNA-gate and toward the C-gate, the upper portion of DNA-gate becomes narrower, and the lower portion becomes wider. At the same time, the central cavity, enclosed by the helix bundles connecting the DNA- and the C-gate, becomes wider and flatter, creating the "squeezed" appearance in conformation 3. However, the conformation of each half-DNA gate changes little, relative to the crystal structure, as shown by the newly added RMSD plots in Supplementary Figure 8a; thus, we are seeing mainly a rearrangement of the two protomers' relative position and orientation, as now stated in the revised text (Supplementary Figure 8, caption). Furthermore, as mentioned in the original and revised text (shown below), this structure resembles a previously reported crystal structure of Bacillus gyrase, as shown in Supplementary Fig. 8b and cited in reference #40. Thus, we would argue that the conformation in step 3 is plausible.

Page 11, lines 312 ~ 316: "While the quaternary structure of the OC state appears quite different from the ON state, the conformation of each half-DNA gate in the former changes little relative to the latter, as shown by the RMSD plots (Supplementary Fig. 8a). This indicates we are seeing mainly a rearrangement of the two protomers' relative position and orientation."

[The reported structure is of the beta isoform of TopoII. The authors report (abstract and page 9, lines 245-256) that the new structure in part explains why topo2 preferentially relaxes positive supercoils. However, as shown by Osheroff and co-workers (Reference 32), the alpha isoform shows a preference for positive supercoils. On the other hand, it seems that the beta isoform (the structure reported in this paper) shows no such preference. Further, Timsit reported (Nucleic Acids Res. 2011 Nov; 39(20): 8665 - 8676) that for the initial clamping of the G- and T-segments, right-handed crosses are optimal. For the T-segment expulsion, repulsive interactions between the DNA segments were proposed in a left-handed cross-over. Any comments on this?]

We thank the Reviewer for directing us to the very interesting work from Prof. Timsit. After carefully reading the papers from Timsit (NAR 2011 **39**: 8665–76), Cozzarelli (PNAS 2003 **100**: 8654–59), Croquette (PNAS 2003 **100**: 9820–25), and Berger (JMB 2005 **351**: 545–61), we concluded that the controversy pointed out by the reviewer was resulted from the use of

seemingly contrastive terms for describing the handedness of DNA crossings. Specifically, the DNA crossing geometry seen in the (+) supercoiled DNA molecule was referred to as “right-handed crossover” by Timsit (see Fig. 2, NAR 2011 **39**: 8665–76), but was called “left-handed braid (L-braid)”, “left-handed superhelix”, or “superhelix with left-handed configuration” by Cozzarelli (see Fig. 1, PNAS 2003 **100**: 8654–59), Croquette (see Fig. 1, PNAS 2003 **100**: 9820–25), and Berger (see Fig. 1, JMB 2005 **351**: 545–61). Since it appears that the term “left-handed” are more commonly used to describe (+) supercoils, we decided to follow this convention in our paper. Nevertheless, to avoid potential confusion regarding this issue, the use of “left-handed DNA crossover” have been replaced by “left-handed DNA braid” in Abstract as well as in text. In addition, we have commented on and put Timsit’s proposal in the context of our new structure (shown below).

Page 9, lines 258 ~ 263: “Given that the formation of the L-braid is energetically more favorable than the R-braid³⁷, the T-segment may spontaneously dock onto the G-segment with a L-braid-like crossing geometry even before the DNA-conducting channel is formed, and the T-segment can instantaneously enter the channel without the need to reorient as soon as the DNA-gate opens.”

The reviewer’s concern regarding whether the human Top2 β exhibits a preference for (+) supercoils is well taken. Our argument is that we thought a small but likely significant preference of Top2 β for (+) supercoils can be recognized in the Figure 6 of the Osheroff’s 2005 JBC paper (reference #35; #32 in the original submission). Given that this so-called “chirality-sensing” is mainly mediated by the C-terminal domain that is diverged between Top2 α and Top2 β , it is reasonable to expect that the preference seen in Top2 β would be much smaller, which is why we say the structural feature observed in the new conformation “**may**” contributes “**in part**” to the distinguish between (+) and (-) supercoils. We will have no objection to take out this speculation if the reviewer feels it is more appropriate to do so.

[Minor Corrections:

- 1. Page 8, Line 212- There is a mention of A’ α 19 in Fig 3a. However, the helix is not marked in the figure.**
- 2. Reference 68 and 69 have to be superscripted**
- 3. From the supplementary Figure 7, I understand that 160 ns long simulations are performed. It should be mentioned in the method section, too.]**

We appreciate the reviewer’s keen attention. All these problems have been fixed.

1. We have marked the helices A’ α 18 and A’ α 19 in Fig. 3a.
2. We have thoroughly checked the reference format.
3. The simulation time is now given in the Method section (page 22: lines 632 ~ 633).

Response to Reviewer #2:

[This is a very interesting structural study in the topo II area as it is a first structure determination of an open G-gate.]

We appreciate this positive assessment of the present contribution.

[As the authors correctly state all the previous structures are for closed G-gates and hence do not address this question of how the T-segment DNA traverses the open G-gate although there has been based on molecular mechanics modelling a view of the trajectory and how this may occur in the 2nd movie in Laponogov et al. Nucleic Acids Research, Volume 41, Issue 21, 1 November 2013, Pages 9911–9923, <https://doi.org/10.1093/nar/gkt749>.]

We appreciate the reminder of the movie contributed by Laponogov and co-workers, and now cite it in relation to the open-book picture of strand passage through the DNA gate (reference #16). It is perhaps worth noting that this prior movie is not based on molecular simulations, but on a manual modeling procedure.

[The X-ray crystallography has been very carefully executed with iodo-U substituted DNAs use to unambiguously determine the positioning of the DNA and hence the veracity of the structures. The refinement has also been very carefully performed and the structures are well refined.]

We appreciate this positive assessment on the quality of the reported structures.

[I could see the crystallographic parameters for the P3sub2 structure in the supplementary, but not that for the P2sub1 structure talked about in the text but the crystallographic parameters are not tabulated. Also in the text the space group is referred to as P3 and not P3sub2! This should be corrected as P3 is also a possible space group (also the Laue group) but the systematic absences and MR provide the definitive space group as being P3sub2.]

We thank the reviewer for pointing out these problems. Indeed, the P2sub1 structure and a second P3sub221 structure, determined using the 5-iododeoxyuridine-labeled DNA, were used in this study as a part of the effort to verify the formation of Top2 cleavage complex. The atomic coordinates and structure factors of these two structures have been deposited in the Protein Data Bank (PDBid: 5ZQF and 5ZRF). The data collection and refinement statistics for

these two structures are reported in the revised Supplementary Table 1.

We apologize for not identifying the space group properly in the initial submission. The correct space group should be P3sub221. We have fixed this problem in the revised manuscript.

[The steered molecular dynamics simulations are very interesting but I am very surprised that a pair of the helices forming the C-gate completely unwrap at the end of the simulation. I have run molecular dynamics on systems although not on these topo II-DNA and topo II-DNA-drug systems, but to me this would be intuitively unlikely as these are very long helices stabilised by many H bonds between the carbonyls and the main chain NHs all along the length of the helices. Are the force field constraints being correctly applied at these ending stages? Is this a true reflection of what happens in solution, this unwrapping on these time scales?]

The Reviewer's points are well taken. Our simulations were carried out using force field parameters described in a prior study (Nucleic Acids Res 2015, 43(14): 6772–6786). An unbiased microsecond molecular dynamics simulation was presented in that work, and these alpha-helices remained well folded. In the steered simulations presented in the current study, we applied positional and distance restraints to accelerate the strand passage process, which probably occurs on a considerably longer time-scale in nature. We did not apply additional constraints to the C-gate helices, because we aimed to see how the protein's conformation would respond to complete strand passage with minimal added perturbations. We are not aware of experimental evidence for or against the unwinding of these helices during the catalytic cycle. However, there is precedent for local helix unwinding in a DNA-binding enzyme, as the central bridge helix of RNA polymerase I (Pol I) can adopt either a fully folded conformation, in the elongation complex, or an unwound state, in the free enzyme (Nature. 540:607–610, 2016.) This point is now made in the revised Discussion (shown below).

Page 14, line 408 ~ page 15, line 414: “Comparing the conformations of the ON and the OC states, one may notice that the coiled-coil region underwent a helix-to-coil transition in the latter. We are not aware of experimental evidence for or against the unwinding of these helices during the catalytic cycle of Top2. However, there are precedents for functionally relevant local helix unwinding events. For example, the central bridge helix of RNA polymerase I can adopt either a fully folded conformation, in the elongation complex, or an unwound state, in the free enzyme⁵⁰.”

[I think that this paper is an extremely important contribution to the field of

topoisomerase structure and mechanism]

We appreciate this positive assessment of the present contribution.

REVIEWERS' COMMENTS:

Reviewer #1 (Remarks to the Author):

The authors have done a great job to address all my concerns. This manuscript addresses important aspects related to topo2 function, while opening to the possibility of further mechanistic studies on Topo2. I have no further comments, and I congratulate with the authors for the excellent work and relevant findings.